# The Clinical Utility of lncRNAs and Their Application as Molecular Biomarkers in Breast Cancer

**DOI:** 10.3390/ijms24087426

**Published:** 2023-04-18

**Authors:** Cristian Arriaga-Canon, Laura Contreras-Espinosa, Sergio Aguilar-Villanueva, Enrique Bargalló-Rocha, José Antonio García-Gordillo, Paula Cabrera-Galeana, Clementina Castro-Hernández, Francisco Jiménez-Trejo, L. A. Herrera

**Affiliations:** 1Unidad de Investigación Biomédica en Cáncer, Instituto Nacional de Cancerología-Instituto de Investigaciones Biomédicas, UNAM, Avenida San Fernando No. 22 Col. Sección XVI, Tlalpan, Mexico City C.P. 14080, Mexico; 2Posgrado en Ciencias Biológicas, Unidad de Posgrado, Edificio D, 1° Piso, Circuito de Posgrados, Ciudad Universitaria, Coyoacán, Mexico City C.P. 04510, Mexico; 3Departamento de Tumores Mamarios, Instituto Nacional de Cancerología, Tlalpan, Mexico City C.P. 14080, Mexico; 4Departamento de Oncología Médica de Mama, Instituto Nacional de Cancerología, Tlalpan, Mexico City C.P. 14080, Mexico; 5Instituto Nacional de Pediatría, Coyoacán, Mexico City C.P. 04530, Mexico; 6Tecnológico de Monterrey, Escuela de Medicina y Ciencias de la Salud, Monterrey C.P. 64710, Mexico

**Keywords:** lncRNA, breast cancer, biomarkers

## Abstract

Given their tumor-specific and stage-specific gene expression, long non-coding RNAs (lncRNAs) have demonstrated to be potential molecular biomarkers for diagnosis, prognosis, and treatment response. Particularly, the lncRNAs *DSCAM-AS1* and *GATA3-AS1* serve as examples of this because of their high subtype-specific expression profile in luminal B-like breast cancer. This makes them candidates to use as molecular biomarkers in clinical practice. However, lncRNA studies in breast cancer are limited in sample size and are restricted to the determination of their biological function, which represents an obstacle for its inclusion as molecular biomarkers of clinical utility. Nevertheless, due to their expression specificity among diseases, such as cancer, and their stability in body fluids, lncRNAs are promising molecular biomarkers that could improve the reliability, sensitivity, and specificity of molecular techniques used in clinical diagnosis. The development of lncRNA-based diagnostics and lncRNA-based therapeutics will be useful in routine medical practice to improve patient clinical management and quality of life.

## 1. Introduction

Long non-coding RNAs (lncRNAs) are known to be tissue [1], cancer specific [2], and disease specific in breast cancer [3]. In particular, their expression has also been reported to be distinct under several conditions, for example, in cancer progression or in disease resistant to systemic treatments [4]. Currently, several examples of lncRNAs have been described as potential clinical biomarkers for predicting response to therapy or for prognosis in breast cancer, such as *HOTAIR*, *H19*, and *DSCAM-AS1. HOTAIR*, a −2 kb lncRNA located in chromosome 12, regulates gene expression by recruiting PRC2 and LSD1 complexes to their target gene regions, modulating the enrichment of H3K27 and H3K4 methylation [5], and it has also been described that *HOTAIR* overexpression increases cell proliferation by an enhanced estrogen receptor signaling pathway [6]. *H19* is also a −2 kb lncRNA, located in chromosome 11, and evidence suggests that it modulates gene expression by the miRNA sponge mechanism [7]. These two lncRNAs, *HOTAIR* and *H19*, have been demonstrated to be overexpressed in breast cancer; this has been related with predictive value to assess resistance to neoadjuvant chemotherapy in breast cancer [8,9]. *HOTAIR* has been particularly associated with metastasis in breast cancer [10]. Additionally, *DSCAM-AS1* is another lncRNA of −1 kb of length, located in chromosome 21, whose molecular mechanism is still unknown, but several studies suggest that it has a miRNA sponge function, particularly in carcinogenesis [11], and its silencing induces a reduction in proliferation in breast cancer cell lines [12]; its overexpression has been associated with resistance to tamoxifen therapy [13]. Additionally, *GATA3-AS1* was first identified as a lncRNA, differentially expressed in breast cancer patients by Zhang, et al. [14]. This lncRNA of ~2 kb of length is in chromosome 10, and it has also been associated with Th2 lymphocytes differentiation [15] by regulating its adjacent gene *GATA3*, a transcriptional factor involved in immune response [16] and breast cancer [17]. The mechanisms by which *GATA3-AS1* regulates the function of *GATA3* are not fully understood, but Gibbons et al. described a molecular mechanism in human T lymphocytes in which *GATA3-AS1* forms an R-Loop in *GATA3* locus; this structure allows for the recruitment of a chromatin remodeler, such as MLL, leading to the activation of *GATA3* transcription [16]. Because of its expression in breast cancer, *GATA3-AS1* has also been proposed as a potential biomarker of response to neoadjuvant chemotherapy in patients with locally advanced breast cancer of Luminal B subtype [17]; it is also related to the immune response since it regulates the differentiation of Th2 lymphocytes, as mentioned before [16]. Furthermore, Zhang et al. reported that its expression is related to the immune response of mammary tumors by stabilizing protein levels of PD-L1 in triple negative breast cancer [18]. Taken together, these results suggest that *GATA3-AS1* could be used in clinical practice as a biomarker that would form a part of targeted therapy strategies for PD-L1 positive tumors in triple negative breast cancer, for which pertuzumab, atezolizumab, and nab-paclitaxel are implemented as treatments [19]. Thus, lncRNAs are associated with several clinical conditions, such as resistance to therapy or metastasis development in breast cancer. This gives the possibility to propose them as novel molecular biomarkers for clinical practice. However, as is the case for many lncRNAs, the research to determine their clinical application is limited by several factors, including study design and the number of samples used to determine the clinical utility of lncRNAs.

## 2. Sample Size Calculation for Biomarker Discovery in Clinical Research

Currently, it is necessary to direct research efforts towards the use of lncRNAs in targeted therapies with sample sizes that enable adequate sensitivity and specificity [20]. For example, the studies performed by Contreras-Espinosa et al. and other research works in translational medicine, in which they identified the lncRNAs *GATA3-AS1* [17], *LINC02544* [21], and *H19* [9], are related to neoadjuvant chemotherapy response in luminal breast cancer phenotypes, while lncRNA *AC009283.1* was associated with tumorigenic cell signaling pathways in HER-2 enriched breast cancer phenotypes [22]; together, these results demonstrated the association of lncRNAs with different clinical characteristics in breast cancer. However, a common characteristic of these studies is the small sample size. For the identification of a molecule as a novel biomarker, like a lncRNA, the study population should be as large as the study needs to obtain from it a reliable result about lncRNAs to be candidates for biomarker development, which means that the study must be planned with adequate statistical criteria, considering the lowest possible *p*-value (*p*-value < 0.05), which would allow obtaining lncRNAs as biomarker candidates. To identify the greatest potential for predictive biomarkers with clinical applicability, several requirements have been described previously by Pepe et al. who designed the prospective-specimen-collection, retrospective-blinded-evaluation (ProBE) for selecting samples [23], including specific criteria, such as the definition of biomarker performance and the proportion of useful markers that the study should identify and the tolerable number of useless markers. This represents the guidelines that are now applied routinely in validation research, but not in discovery research, and they could help improve the quality, as well as usefulness, in lncRNA biomarker discovery.

To achieve this goal, the sample size must be properly determined depending on the phase of the study and the type of study to be carried out [24]. For the discovery phase in preclinical studies for biomarker discovery, based on RNA expression, it is important to consider implementing the Next Generation Sequencing approach, due to its ability to analyze the entire transcriptome through the RNA-Seq technique [25]. Particularly, in the discovery phase for preclinical studies based on RNA-Seq data analysis, the sample size can be small (*n* > 3) because the statistical power of the study relies on the type of transcript of interest (protein coding genes or ncRNAs) and in the reduction of the dispersion generated by the biological heterogeneity [26,27]. The principal parameters that can be used to improve statistical power are the design of the library (a paired-end library enhances more statistical power compared to single-end libraries), the design of the study (a tissue level expression analysis decreases dispersion compared to population study), and the depth of the sequencing (a study with a depth greater than 25 million reads increases the statistical power) [26]. However, it is desirable to have at least three biological replicates for each biological condition to be analyzed, for example, for the discovery of predictive biomarkers, three samples per each responder patient and three samples per each non-responder patient should be considered in the experimental design of a case-control study (*n* = 6) [28]. Additionally, it is necessary to determine the data distribution for subsequent statistical analysis, which includes the use of normal distribution tests, such as Kolmogorov-Smirnov and Shapiro-Wilk tests [29]. If data is normally distributed, then parametric tests, such as *t*-test, ANOVA test, linear regression, and Pearson correlation test, are implemented to determine the clinical association of the candidate lncRNA [30]. If data follows a non-normal distribution, then non-parametric tests, such as Fisher’s exact test, Mann-Whitney U test, and Spearman correlation test, are implemented to determine these associations [31], which should be validated in a larger and independent cohort of patients [32].

For the validation phase, it is assumed that the results of the discovery phase will be replicated, and to achieve this, we must consider fitting the sample size of the replication phase based on the conditional power calculation, which is the probability that the replication study leads to a statistically significant conclusion, as is described by Micheloud and Held [28]. Once the candidates for use as predictive biomarkers are selected, the results should be validated in larger cohorts in the Replication Phase, considering only the expression levels of candidate RNAs and their predictive value. This must be done by determining the data distribution, calculating sensitivity, and specificity for each candidate or for the whole set, which leads to the development of a molecular signature, which assumes normal distribution [33,34]. The aim of this method is to determine the true positives plus false negatives value based on the pre-established distribution value (Z = 1.96), sensitivity, specificity, and the maximum confidence interval of 95% (W = 0.1). For biomarkers, in general, the minimum acceptable values for sensitivity and specificity are 0.8 (80%) and 0.6 (60%), respectively [35]; thus, it is desirable to set these values for further calculations and determination of the optimal sample size (*N*). Finally, each sample size (sensitivity and specificity) is added to obtain the total sample size required for the study [33], which is statistically optimal for obtaining reliable information to determine the utility of the candidate biomarker in clinical practice. Here, we present a compilation of the scientific literature dealing with lncRNAs, and their clinical utility in breast cancer is presented, for which we performed an exhaustive search conducted by Pubmed, Google Scholar, and Connected Papers using keywords relating to lncRNAs and their clinical application in breast cancer, which includes “biomarkers”, “non-coding RNAs”, “lncRNAs”, “breast cancer”, “prognosis”, “prediction”, “RNA-Seq”, “transcriptome”, “sample size”, and “clinical trials”.

### Sample Size Determination for lncRNA Studies in Breast Cancer Research

One example of optimal sample size is the study of Niknafs et al., in which they included 946 patients for the discovery phase and 758 patients for the validation phase. They described the usefulness of the lncRNA *DSCAM-AS1* as a biomarker of resistance to tamoxifen in endocrine therapy because of its overexpression in patients with estrogen receptor positive breast cancer [13], as it is also the case for *GATA3-AS1* [17], for which the accurate sample size for the replication phase should be 1295 patients [33]. Despite this, the use of a convenient sample size in studies for lncRNA biomarker discovery is also relevant for the identification of predictive biomarkers because several studies have identified predictive biomarkers in neoadjuvant chemotherapy, such as *H19* [9], *MALAT1* [36], and *LINC02544* [21], which, despite the reduced sample size, showed a potential utility in clinical application. Additionally, public databases in which the results of the expression analyses are stored by massive parallel sequencing of coding genes, such as *GEPIA* [37], as well as non-coding genes, such as *TANRIC* [38], both corresponding to the expression profiles of patients found in the Cancer Genome Atlas (TCGA) cohorts, validates the results obtained in studies such as those mentioned above, in which it was not possible to obtain more samples. In summary, the importance of properly implementing the design of a clinical study ensures the reliability of the obtained data in the results, and therefore, it could be validated in a replication phase to determine the clinical value of a biomarker based on lncRNA.

## 3. The Current Use of lncRNAs as Clinical Biomarkers in Clinical Practice

Although their usefulness in clinical practice is poorly understood, the use of lncRNAs as predictive biomarkers in response to therapy has advantages compared to protein-based and mRNA-based biomarkers [39] since they present tissue and stage specific expression [40]; this gives them greater sensitivity and specificity [41], particularly in tumors with hormone sensitivity, such as the prostatic adenocarcinoma, in which some lncRNAs with clinical utility, such as *SChLAP1* [42], *lncRNA-p21* [43], and *PCA3* [44], have been identified. As their association with prostate cancer has already been established, this allows for their use in clinical practice. For example, *SChLAP1* is a lncRNA whose length is 854 nt, is transcribed from chromosome 2, and is differentially expressed in bladder normal tissue and prostate cancer tissue. It was first identified in paraffin-embedded tissue biopsies by in situ hybridization (ISH). The biological function of *SChLAP1* is related to the regulation of the SWI/SNF chromatin-modifying complex; this lncRNA antagonizes the genome-wide localization of this protein complex, which is related to the promotion of invasiveness and metastasis in LNCaP and 22Rv1, as well as in Du145 cancer cell lines [45]. Moreover, *SChLAP1* expression has also been associated with metastasis (odds ratio [OR] 2.45, 95% CI 1.70–3.53; *p*-value < 0·0001) and cancer progression (hazard ratio = 1.99, *p*-value = 0.032) in prostate cancer patients [46]. Additionally, *lincRNA-p21*, which is a lncRNA, has been shown to be differentially expressed in prostate cancer [47]; its biological function is principally the regulation of apoptosis, cell proliferation [48], and cell cycle by its interaction with MDM2 and STAT3 [47]. *lincRNA-p21* has also been related to disease progression in prostate cancer in preclinical studies, as its overexpression in castration-resistant patients who were treated with enzalutamide is associated with less overall survival (*p*-value = 0.04), which indicates that *lincRNA-p21* could also be a useful predictive biomarker for enzalutamide treatment [47]. Finally, the Prostate Cancer Antigen 3 (*PCA3*), a lncRNA of 3 Kb in length transcribed in chromosome 9, is present in prostate cancer with high tissue-expression specificity, described first by Bussemakers et al. in 1999 [49]. Currently, *PCA3* is also an auxiliary biomarker in prostate cancer; its use was approved by the Food and Drug Administration (FDA) in 2012 [44] due to its clinical utility by reducing the number of unnecessary biopsies in patients. Additionally, *PCA3* has been reported to be related to the survival of prostate tumor cells by regulating the androgen receptor signaling pathway, as well as regulating the epithelial-mesenchymal transition (EMT) by modulating some targets, such as E-cadherin and TWIST [50,51]. Furthermore, it has been used in gene signature PROGENSA to determine which patients with a previous negative biopsy [52] need a second biopsy [53]. As described above, the use of molecular biomarkers based on lncRNA expression for prostate cancer has demonstrated the utility of this RNA biotype in clinical practice. Likewise, this could be extended to breast cancer clinical application since both carcinomas are characterized as hormone-sensitive [54], and there is experimental evidence of lncRNA expression related with clinical outcome, such as lincRNA-ROR, in which PCA3 regulates EMT by modulating E-cadherin functions [55]. Thus, it is necessary to implement more research to have similar results in biomarker discovery for breast cancer.

Moreover, in prostate cancer research, it has been established that, although the lncRNA expression itself has clinical utility, the identification and detection of different biotypes, such as mRNAs, and genetic fusions also has utility in clinical practice [56]. Indeed, there are reports in scientific literature that demonstrate that the combination of lncRNA, mRNA, and genetic fusions in molecular signatures has improved the sensitivity or specificity of assays based priorly only in the expression of one gene [57]. One example is PROGENSA, which is based on *PCA3* expression and is associated with a sensitivity of 66–72% and a specificity of 58–76% [58,59], while Mi Prostate Score, an urinary test based on the detection of PSA (mRNA), *PCA3* (lncRNA), and TMPRSS2-ERG (genetic fusion) [60], has an associated sensitivity value of 95%. This demonstrates that the combinatorial use of mRNAs, lncRNAs, and genetic fusions can improve the results of laboratory tests for prostate cancer, and this could be extended to breast cancer research.

### 3.1. Challenges and Perspectives for lncRNA Clinical Application as Predictive Biomarkers for Breast Cancer Management

For breast cancer, there are few studies that support the use of lncRNAs or the combination with other biotypes as molecular predictive or prognostic biomarkers in clinical practice, and none of them have been approved for commercial distribution in prostate cancer, as in the case of PROGENSA, although there is already evidence in scientific literature about their potential as biomarkers in decision-making for the management of breast cancer patients [61,62,63]. The best example to describe the potential clinical utility of a lncRNA in patients with breast cancer is the study performed by Berger et al. in which the existence of lncRNA-coding gene regulation networks, such as *NEAT1*, *TERC,* and *TUG1*, together with other mRNAs, such as ESR1, AR, and SOX2, make it possible to classify patients with gynecological cancers and breast cancer into 6 clusters, which are related directly to their phenotypes and mainly to the immune response, as well as to the expression of hormone receptors in patients particularly associated with the estrogen receptor signaling pathway. This biomarker can be used for diagnostic, predictive, and prognostic purposes in breast cancer patients [64]. Furthermore, this has also been demonstrated by Niknafs et al., who described the use of *DSCAM-AS1* expression as part of the characteristics of luminal tumors that are positive to hormone receptor expression [13]. It has also been described by Contreras-Espinosa et al. for *GATA3-AS1* [17] and for the LINC01087, which expression profile is also related with luminal phenotypes in breast cancer [65]. This suggests that *GATA3-AS1* expression may be a relevant molecular characteristic that defines luminal tumors [66]. However, there are additional emerging lncRNAs that have been described as potential biomarkers in cancer, such as *HOTAIR*, *DSCAM-AS1*, and *GATA3-AS1* in breast cancer [8,13,17], *MALAT1* in lung cancer [67], *H19* in colorectal cancer [68], *HULC* in liver cancer [69], *UCA1* in bladder cancer [70], and *DLEU1* in endometrial cancer [71]. Among other lincRNAs [72], the applicability of lncRNAs in the molecular diagnostic area and their use in laboratory tests for clinical diagnosis in the near future largely depends on the expansion of knowledge about their association with different clinical variables, such as response to treatment and overall survival, as well as their inclusion in clinical trials in order to determine and validate the benefits of their use in clinical routine, as it has been made for coding genes before [73]. Thus, there are still many studies to be carried out in order to include lncRNAs more frequently in laboratory tests for the patient’s workup and treatment, not only in oncology, but also for other pathologies, like cardiovascular diseases [74] and diabetes [75], which are also leading causes of morbidity worldwide [76]. Taken together, these results suggest that lncRNAs may be relevant biomolecules that could allow oncologists to differentiate patients who do not respond to therapy, regardless of the molecular heterogeneity of breast tumors, which represents an important challenge in oncology practice [77].

### 3.2. The Use of lncRNAs as Molecular Biomarkers in the RNA-Based Therapeutics Era

A molecular feature advantage that distinguishes lncRNAs is their stability in biological samples, such as blood, urine, or saliva (median half-life ~3.5 h) [78]. This is due to their transport in exosomes, microvesicles, apoptotic bodies, high density lipoprotein capsules, or into circulating tumor cells [79], in contrast with mRNAs, which are characterized by their instability in body fluids (median half-life < 2 h) [80]. This allows the detection of lncRNAs by non-invasive techniques through the use of liquid biopsies, such as urine and saliva, and less-invasive methods, such as serum and plasma [40], as has been reported for lncRNAs *HOTAIR* [8] and *H19* [9] in breast cancer, as well as for *MALAT1*, which has been shown to be a serological marker in breast cancer [36] and a diagnostic biomarker for oral squamous cell carcinoma that can be detected by saliva testing [81]. The detection of these lncRNA is performed by a quantitative polymerase chain reaction (qPCR) in RNA extracted from serum or saliva obtained from patients. Likewise, it is possible to detect lncRNAs with the use of other techniques with higher sensitivity, such as ISH-RNA, which has been used for the rapid detection of markers, such as HER2 in breast cancer, [82] with greater sensitivity and specificity (99% and 98%, respectively) when compared to HER2 immunohistochemistry (IHC) assay detection (95% and 98%, respectively) [83]. The ISH-RNA assay has also allowed the detection of lncRNA *SNHG3* as a potential diagnostic biomarker, distinguishing between normal breast tissue and cancerous breast tissues [84]. Furthermore, there are novel molecular approaches, such as spatial transcriptomics, which allow for the identification of a signature based on 798 transcripts, including the lncRNA *LINC00657*, that could be implemented in machine learning methods to distinguish invasive breast cancer [62]. In summary, the implementation of molecular biology techniques for lncRNA-based biomarkers detection in clinical practice could improve the reliability of the results of laboratory tests and the accuracy of oncological diagnosis.

As discussed above, the implementation of *PCA3*, *DSCAM-AS1*, and *GATA3-AS1* as other lncRNA molecular biomarkers represents a novel approach for the clinical management of the oncological patient (Figure 1) since their expression is tissue-specific, disease-specific, and is associated to a particular stability in body fluids [2], contributing to the development of precision medicine; this is because lncRNA-based biomarkers offer simple and reliable tests [85]. Altogether, this represents the lncRNA-based diagnostics [86,87], a new concept in medicine which integrates the potential use of lncRNAs as molecular biomarkers, with application in clinical practice, that will improve patient management in three main aspects. The first is the use of non-invasive techniques for laboratory tests (e.g., liquid biopsies); this has proven to be useful in clinical routines as the urine analysis, which is currently in practice with the use of *PCA3* [88]. The implementation of these molecular assays, with fluids like urine and saliva, have the main objective of benefitting patient management because these methods allow the oncologists to perform the diagnostic and follow up of patients in less invasive manners, with the accuracy improved, due to the capability of these non-invasive methods to avoid some bias, like tumor cell heterogeneity [89]. Hence, the detection of lncRNAs by the implementation of non-invasive methods, such as urine and saliva analysis, is a promising improvement in clinical routine. Second, the use of time and cost-efficient detection techniques, such as qPCR, which take ~2 h to get results [90] in contrast to IHC, which takes approximately 2 days or more [91], will directly impact the optimization of the oncologist decision making, for example, in the decision for treatment selection for breast cancer patients. Third, the improvement in result accuracy for laboratory tests. Because of the high specific expression profile of lncRNAs, as well as their sensibility and specificity, differential diagnosis and early diagnosis are easier and can also be combined with pathological imaging processing that involves the use of X-ray imaging, magnetic resonance imaging, nuclear medicine imaging, and ultrasound imaging, which are techniques with routinary use in clinical practice [92]. Thus, the combinatorial use of molecular and image biomarkers could lead to the improvement in diagnosis, prediction, and prognosis values [93], which have been demonstrated by the implementation of machine learning algorithms for the integration of molecular imaging and clinical data [94,95,96]. However, to achieve the implementation of combinatorial biomarkers in breast cancer, the development of appropriate research protocols is necessary to demonstrate and validate their usefulness in clinical practice.

### 3.3. The Current Challenges for lncRNA Research and for Their Implementation as Molecular Biomarkers in Routine Clinical Practice

Finally, it should be noted that the principal objectives in the investigation of the use of lncRNAs as current molecular biomarkers in breast cancer have mostly been aimed at determining the biological function of lncRNA [100,101] or their ability to describe mammary tumors molecularly, as is the case of lncRNA *EPIC1* described by Wang et al., which has been identified as an oncogene in breast cancer that promotes cell cycle and has been associated with poor overall survival (hazard ratio ~2, *p*-value = 0.005) [99]. However, it has not yet been possible to apply this knowledge in biomarker development for routine used in clinical practice, as it occurs with *PCA3* in prostate cancer [44]. This happens because of three main reasons: (1) the sample size used for the discovery and validation of these potential biomarkers, (2) the lack of clinical trials focused on exploring the association of lncRNAs with clinical variables, and (3) the lack of clarity and accurate use of clinical definitions that involve biomarkers and clinical assessments, which make the development of new clinical tools based on novel molecular markers, such as lncRNAs, and their inclusion in clinical practice difficult [102]. However, their advantages over other biomolecules, such as proteins and mRNAs, have been demonstrated [40], as is the case of *GATA3-AS1*, which predicts neoadjuvant chemotherapy resistance in luminal B-like breast cancer patients with a sensitivity of 92% and specificity of 75% (*p*-value = 0.0001) [17] compared to Ki-67, a clinical biomarker for neoadjuvant chemotherapy response prediction in breast cancer (sensitivity: 95.7%, specificity: 54.3%, *p*-value  =  0.002) [103]. Another example is *H19* [68,104] and *DSCAM-AS1* [13,105]; not only has their biological function been described, but so has their applicability in clinical practice by establishing associations with clinical variables, such as estrogen receptor expression, which has potential application for diagnosis (sensitivity, 100.0%; specificity, 97.0%; *p*-value < 0.001), as well as predictive and prognostic features [105]. Furthermore, there are also studies for lncRNAs that are used as genetic signatures [4,14,106]. Wang et al. reported the use of a gene signature based on lncRNA expression that included *NEAT1*, and its predictive value for neoadjuvant chemotherapy response was described (sensitivity, 69.9%; specificity, 77.8%; *p*-value < 0.0001) [106]. However, the sample size in these studies is small and this has not allowed for the scale up of the applications of lncRNAs to the dimensions of a clinical trial, as is the case of *H19* [9], *HOTAIR* [8], *MALAT1* [36], and *GATA3-AS1* [17], which have predictive value for neoadjuvant chemotherapy response. Furthermore, in general, there are few studies in the clinical setting that investigate the potential use of lncRNAs as diagnostic, predictive, or prognostic biomarkers in clinical practice for cancer (Appendix A) since there are only a few studies in preclinical stages and only three clinical trials that include lncRNAs as biomarkers developed specifically for breast cancer research (Table 1) and are not specifically dedicated to a particular lncRNA or do not yet have a publication related to lncRNA application [107,108,109]. This is despite the evidence for their roles as regulators of cell proliferation and survival in mammary tumors (Table 1), which represents a disadvantage in the development of lncRNA-based biomarkers in breast cancer, particularly in locally advanced stages, and makes their inclusion to molecular signatures harder, even if their utility could improve the sensibility and specificity of molecular biomarkers based on coding genes. This has been demonstrated for *H19*, which has a sensitivity of 56% in the identification of metastatic breast cancer [110] compared with CA-15-3, a tumor marker with a sensitivity of 42% in the same prognostic application [111]. This is evidence that lncRNA inclusion enhances gene signatures for clinical trials.

Practice, as is shown by Liu et al., with the comparison of 5-gene signature integrated by 4 mRNAs (OR7C1, TBX2, RSPH4A, and C2orf61) and the lncRNA *AC10538*, with 21-gene signature Oncotype Dx, in predicting overall survival in early-stage breast cancer patients showed that a 5-gene signature performed better in prognosis value than 21-gene in a TCGA cohort (areas under curves 0.807 and 0.572, respectively) [121]. This demonstrates that the integration of lncRNA to an mRNA signature could improve the prognosis value, as was also demonstrated for the combinatorial use of lncRNA *MALAT1* and Oncotype Dx, for predicting the diagnosis of early-stage breast cancer [122].

Similar results have been reported for Acute Lymphoblastic Leukemia, in which a 7-gene signature, including one lncRNA, *LINC00652*, and six mRNAs, INSL3, NIPAL2, REN, RIMS2, RPRM, and SNAP91, had an area under the curve of 0.9 for predicting 5-year overall survival [123]. Moreover, Zhou and collaborators described a 12 lncRNA gene expression panel that is capable of predicting the risk of recurrence in breast cancer patients [124]. Another example is the study performed by Shen and collaborators, in which they demonstrated that a 11-lncRNA gene signature, that has an associated prognostic value (HR = 1.328 in univariate Cox regression analysis, with *p*-value < 0.001 and HR = 1.266 in multivariate Cox regression analysis, with *p*-value < 0.001), is also related with immune cell infiltration in breast tumors, which could have a clinical application for breast cancer immunotherapy [125]. Additional studies of the identification of molecular signature biomarkers based on lncRNA expression profiles are listed in Appendix A. This evidence suggests the utility of lncRNAs, in clinical practice, as molecular biomarkers of treatment response prediction and prognosis in cancer and their potential inclusion in already used molecular signatures, like Oncotype Dx [126]. The usefulness of lncRNAs in breast cancer, particularly for locally advanced stages, can therefore be more than just identifying biological properties and treatment response to systemic therapy [39], as is the case of *HOTAIR*, which is related to lymph node metastasis [10] and *HUMT* (*p*-value < 0.001) [127].

As well as the above examples, there are numerous examples of other lncRNAs that have shown potential as biomarkers for predicting the response to treatment in breast cancer in studies conducted in mouse animal models at the preclinical level (Table 2). However, as discussed above, efforts need to be directed to move this research to phase I clinical studies, not only for response prediction biomarkers, but also to biomarkers with diagnostic and prognostic applications in order to determine the benefits that they can bring to the management of oncology patients, including in other areas, such as surgery.

Additionally, in surgical practice, they could help in predicting lymph node burden and, therefore, avoid extensive axillary surgery in patients eligible for genomic tests [151], as it has been done recently with epigenetic signatures [152]. They could also work as biomarkers to predict objective response rates (ORR) to specific therapies in the neoadjuvant setting so tumor downstaging could be more precisely inferred [153]. In metastatic disease, these molecules can also help us to identify progression free survival in specific settings [154], such as oligometastatic disease in luminal-like tumors [155]. In the neoadjuvant setting, differential expression of lncRNAs could help to identify the probability of achieving low residual cancer burden (RCB) [156], and perhaps even its expression in residual disease might identify luminal-like patients that could benefit from adjuvant therapies if their absence/presence is associated with recurrence or death, since pathologic complete response is not an adequate biological surrogate of survival in this molecular subtype [157]. In addition, lncRNAs could have an impact on the development of new therapies based on RNA or RNA-based therapeutics [158] for breast cancer patients. Particularly, lncRNA could be incorporated for immunotherapy purposes [159], for example, as enhancers of a cancer vaccine’s response, as is the case of *LINK-A*, a lncRNA whose inhibition by locked nucleic acids synergistically suppresses tumor progression when it is combined with immune checkpoint blockers (or ICB therapy) [160], which have the same effect that the PVX-410 cancer vaccine [161] has. This in turn suggests that *LINK-A* could be a biomarker for PVX-410 vaccine response or a sensitizer for this vaccine [162]. Another example of the potential use of lncRNAs in cancer therapy is their use as targeted therapies for the treatment of metastatic cancer with the use of systems of release, such as the *MALAT-1* antisense oligonucleotide, stabilized in antisense oligonucleotide-loaded nanostructure, co-functionalized in Au nanoparticles, which reduced metastatic tumors in vivo [163]. Altogether, these results show the potential applicability of lncRNA in different RNA-based therapies, including immunotherapy, which is one of the most promising perspectives for lncRNA therapeutics in cancer.

## 4. Conclusions

The trend of lncRNA research in breast cancer is based on the proposal of a new type of precision medicine that relies on the use of ncRNAs, since it is still in the dark matter of human genome [164] and since its application in the clinical area has not been explored deeply. This is due to the persistence of the exclusively and prioritized persistence of coding genes as biomarkers [165,166,167], although they continue to be of clinical utility when merging with gene signature tests, such as Oncotype Dx, Mammaprint, and Prosigna/PAM50 [168]. It has been shown that they do not represent the complete molecular description of mammary tumors [169], for which it is necessary to integrate the knowledge that research areas, such as genomics, transcriptomics, proteomics, epigenomics, metabolomics, and interactomics, have provided [170,171]. In the future, this will allow for the development of precision medicine for the benefit of cancer patients [97].

Thus, the new era of molecular diagnosis should consider the issues discussed above and, at least, allow for the reflection of three perspectives in its development. First, the integration of lncRNAs in clinical trials for the study and analysis of their potential applicability in clinical diagnosis. Second, the use of lncRNAs as diagnostic molecular biomarkers using non-invasive tests, as is the case for *PCA3* detection in urine and for other promising lncRNAs, such as *HOTAIR* in serological tests and *MALAT-1* in salivary tests, which represent an advance in the proper management of oncology patients, as we discussed before. Third, the integration of lncRNAs in commercial gene signatures and laboratory tests with diagnostic purposes, as was previously discussed, to improve the accuracy and reliability of diagnostic results, which will be reflected in the enhancement of oncology management strategies and in the amelioration of cancer patients’ quality of life. Finally, lncRNAs are part of the RNA world that have potential use as molecular biomarkers, which could be used in the near future as part of routine testing in breast cancer management.

## Figures and Tables

**Figure 1 ijms-24-07426-f001:**
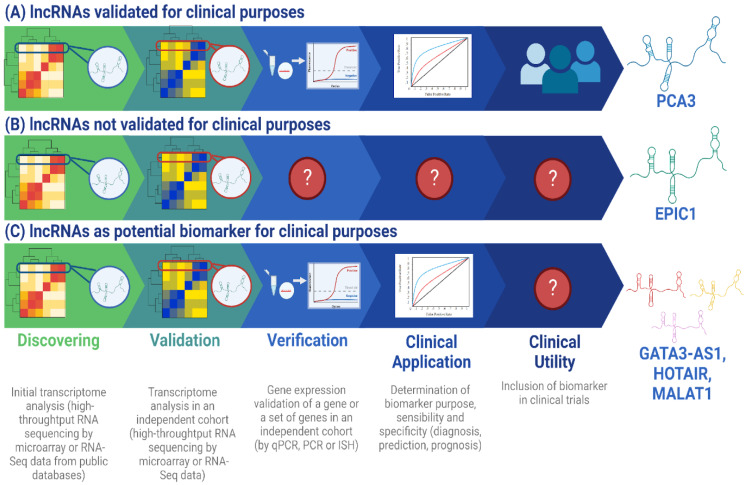
Workflow for lncRNA validation as biomarkers with clinical utility and application. The process of implementing a lncRNA-based biomarker consists of 5 principal steps: discovery, validation, verification, clinical application, and clinical utility [97]. In the discovery step, the objective is to select a lncRNA (or a set) differentially expressed in the condition of interest, like treatment response, and implement it in selected patients. In the validation step, the sample size is increased to determine lncRNA robustness to define the clinical condition of interest and follow the sample size calculation recommended in [20]. In the verification step, lncRNA expression is determined by a clinical laboratory technique, such as qPCR, to verify its viability to be detected in the clinical routine. In the clinical application step, the functionality of the lncRNA as a biomarker for diagnosis, prediction, or prognosis is determined by assessing its sensibility and specificity [20]. Finally, in the clinical utility step, the accuracy of the lncRNA as biomarker is tested in a larger sample size and could be included in clinical trials. (**A**) *PCA3* is an example of a lncRNA that has been validated for clinical application in prostate cancer diagnosis because it represents an FDA-approved lncRNA for clinical purposes. It was discovered from a sample size of 11 patients in the discovery phase [49] and 507 male patients were included in the validation phase in a clinical trial [53]. Additionally, *PCA3* is associated with a sensitivity ranging from 54% to 82% and a specificity range of 56.3% to 89%, which justifies its use in clinical practice [50]. Although *PCA3* was identified by Northern blot technique [49], it has been validated in other studies by high throughput sequencing technologies, which is the principal tool for current biomarker discovery [98]. (**B**) *EPIC1* is a lncRNA that was identified from the analysis of 6475 tumor samples in the discovery phase and 534 samples in the validation phase [99]. However, it has not yet been verified as a biomarker for clinical utility or for clinical application in the prognosis of breast cancer. (**C**) *GATA3-AS1* is a lncRNA proposed as a potential clinical biomarker in predicting treatment response in breast cancer because it has been demonstrated by Zhang et al. that it is overexpressed in breast neoplasia in a differential expression analysis for 85 paired tumor-normal samples and 830 tumor samples in the discovery phase. For the validation step, 50 paired tumor-normal samples and 23 healthy samples were included [14]. Recently, Contreras-Espinosa et al. also identified this lncRNA by a machine learning approach in a sample size of 11 patients for the discovery step and 68 patients for the validation step, which demonstrated its utility as a predictive biomarker [17]. However, it has not been validated for clinical application yet, as other lncRNA which have been proposed as molecular biomarkers for treatment response prediction in breast cancer, like *HOTAIR* [8] and *MALAT1* [36], have not been included in clinical trials for the analysis of their applicability in diagnosis. lncRNA: long non-coding RNA; ISH: in situ hybridization; PCR: Polymerase Chain Reaction; qPCR: quantitative PCR. Created in BioRender.com.

**Table 1 ijms-24-07426-t001:** List of clinical trials and preclinical studies in breast cancer research that also included lncRNAs.

Clinical Trials for lncRNA *					
Clinical Trial ID		Title	Disease	Number of Participants	Phase
NCT02641847		TA(E)C-GP Versus A(E)C-T for the High Risk TNBC Patients and Validation of the mRNA-lncRNA Signature.	Triple Negative Breast Cancer, Breast Cancer.	503	Phase 2 Phase 3
NCT03000764		RNA and Heat Shock Protein Biomarkers in Radiation-induced Fibrosis in Breast Cancer (SPLICI-Rad).	Breast Carcinoma, Fibrosis.	20	Not Applicable
NCT02221999		Weekly Paclitaxel and Cisplatin to Treat Hormone Receptor Positive and Triple Negative Breast Cancer Patients (SHPD002).	Tubular Breast Cancer, Mucinous Breast Cancer, Invasive Ductal Breast Cancer, Inflammatory Breast Cancer.	250	Phase 2 Phase 3
**LncRNA in Preclinical studies**					
**LncRNA**	**Biological Function ^+^**	**Disease**	**Application**		**Reference**
*LINP1*	Proliferation and migration	Triple Negative Breast Cancer	Predictive biomarker for radiotherapy response		[112,113]
*NR2F1-AS1* (*NAS1*)	EMT and invasion	Lung Metastasis Breast Cancer	Prognostic biomarker for metastasis-free survival and relapse-free survival		[114]
*DILA1*	Cell cycle progression and proliferation	ER-Positive Breast Cancer	Predictive biomarker for tamoxifen response. Prognostic biomarker for relapse-free survival		[115]
*LINK-A*	Proliferation and survival	Triple Negative Breast Cancer	Biomarker for stratification of Triple Negative Breast Cancer Patients		[116,117]
*IRENA*	Proliferation	Invasive Breast Carcinoma	Prognostic biomarker. Disease-free survival		[118]
*BDNF-AS*	Proliferation	Triple Negative Breast Cancer	Predictive biomarker for tamoxifen response		[119]
*LINC01271*	Proliferation and migration	Breast Cancer	Metastasis-related biomarker		[120]

* Clinical trials were consulted in https://clinicaltrials.gov/ (accessed on 13 January 2023). Search terms: [LncRNA] AND [Breast Cancer]. ^+^ Studies performed in breast cancer cell lines.

**Table 2 ijms-24-07426-t002:** Studies in mouse models for lncRNA biomarker discovery in breast cancer.

LncRNA	Drug	Expression Status	Response	Reference
*linc00518*	Doxorubicin	Upregulated	Promotes resistance	[128]
*PTENP1*	Doxorubicin	Downregulated	Promotes resistance	[129]
*LOC645166*	Doxorubicin	Upregulated	Promotes resistance	[130]
*UCA1*	Doxorubicin Tamoxifen Paclitaxel	Upregulated	Promotes resistance	[131,132]
*CBR3-AS1*	Doxorubicin	Upregulated	Promotes resistance	[133]
*EGFR-AS1*	Docetaxel	Upregulated	Promotes resistance	[134]
*NONHSAG048143.2*	Docetaxel	Upregulated	Promotes resistance	[135]
*DILA1*	Tamoxifen	Upregulated	Promotes resistance	[115]
*BDNF-AS*	Tamoxifen	Upregulated	Promotes resistance	[119]
*LINP1*	Tamoxifen	Upregulated	Promotes resistance	[112]
*H19*	Tamoxifen Paclitaxel	Upregulated	Promotes resistance	[136,137]
*ATXN8OS*	Tamoxifen	Upregulated	Promotes resistance	[138]
*OIP5-AS1*	Trastuzumab	Upregulated	Promotes resistance	[139]
*ZNF649-AS1*	Trastuzumab	Upregulated	Promotes resistance	[140]
*SNHG7*	Trastuzumab	Upregulated	Promotes resistance	[141]
*LINC00160*	Paclitaxel	Upregulated	Promotes resistance	[142]
*CASC2*	Paclitaxel	Upregulated	Promotes resistance	[143]
*DLX6-AS1*	Cisplatin, Carboplatin	Upregulated	Promotes resistance	[144]
*MIR200CHG*	Cisplatin, Carboplatin	Upregulated	Promotes resistance	[145]
*PRLB*	5-Fluorouracil	Upregulated	Promotes resistance	[146]
*CCAT2*	5-Fluorouracil	Upregulated	Promotes resistance	[147,148]
*SNORD3A*	5-Fluorouracil	Upregulated	Promotes resistance	[149]
*MIAT*	5-Fluorouracil	Upregulated	Promotes resistance	[150]

## Data Availability

Not applicable.

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
