# Peer review of "The Clinical Utility of lncRNAs and Their Application as Molecular Biomarkers in Breast Cancer"

_ijms, 2023, doi:10.3390/ijms24087426_

Round 1
Reviewer 1 Report
Submission ID: ijms-2272191 – " The Clinical Utility of lncRNAs and their Application as Molecular biomarkers in Breast Cancer. "
In this review, Cristian Arriaga-Canon and colleagues report the latest knowledge concerning the role of LncRNAs as molecular biomarkers in breast cancer.
This review is interesting and well written. In my opinion, some few minor points should be addressed:
- Figure 1: are the images in the figure copyrighted? are these pictures public? Are these pictures from a database? probably their origin should be reported in the legend to the figure.
- The authors can improve the Introduction and /or discussion paragraph taking into account several papers: (i) Hou P et al. LincRNA-ROR induces epithelial-to-mesenchymal transition and contributes to breast cancer tumorigenesis and metastasis. Cell Death Dis. 2014; (ii) Zhou et al. Discovery of potential prognostic long non-coding RNA biomarkers for predicting the risk of tumor recurrence of breast cancer patients. Scientific Reports 2016; (iii) Shen et al. Identification and validation of immune-related lncRNA prognostic signature for breast cancer. Genomics 2020; (iii) De Palma et al. The abundance of the long intergenic non-coding RNA 01087 differentiates between luminal and triple-negative breast cancers and predicts patient outcome. Pharmacol Res. 2020; (iv) Bjorklund et al. Subtype and cell type specific expression of lncRNAs provide insight into breast cancer. Communications Biology 2022.
Author Response
Response to Reviewer 1 Comments
Point 1: - Figure 1: are the images in the figure copyrighted? are these pictures public? Are these pictures from a database? probably their origin should be reported in the legend to the figure.
Response 1: Thank you very much for your comment. Figure 1 was designed on the Biorender platform (https://www.biorender.com/). In our case, we pay the annual license for the design of figures of scientific interest. At the end of the footnote in figure 1, it is mentioned that the image was created through biorender (as specified in their terms and conditions of use about this tool). Please see lines 367 and 550.
Point 2: -The authors can improve the Introduction and /or discussion paragraph taking into account several papers: (i) Hou P et al. LincRNA-ROR induces epithelial-to-mesenchymal transition and contributes to breast cancer tumorigenesis and metastasis. Cell Death Dis. 2014; (ii) Zhou et al. Discovery of potential prognostic long non-coding RNA biomarkers for predicting the risk of tumor recurrence of breast cancer patients. Scientific Reports 2016; (iii) Shen et al. Identification and validation of immune-related lncRNA prognostic signature for breast cancer. Genomics 2020; (iii) De Palma et al. The abundance of the long intergenic non-coding RNA 01087 differentiates between luminal and triple-negative breast cancers and predicts patient outcome. Pharmacol Res. 2020; (iv) Bjorklund et al. Subtype and cell type specific expression of lncRNAs provide insight into breast cancer. Communications Biology 2022.
Response 2: Thank you very much for your comment, we agree. In the manuscript, we included the scientific papers about lncRNAs as suggested by reviewer 1. Please see the lines 38, 213-215, 251, 451-452, and 448-455.
In the hope that we have responded to all the suggestions made by the Reviewer#1, we sincerely hope that the new modifications made to the manuscript are in accordance with what was requested. Thank you for the time invested in reviewing our work.

Reviewer 2 Report
No comments. This is a well-written review with excellent coverage of developments and challenges in the field.
Author Response
Response to Reviewer 2 Comments
Point 1: No comments. This is a well-written review with excellent coverage of developments and challenges in the field.
Response 1: Thank you very much for taking the time to read our work. We appreciate the comment made to our review work.

Reviewer 3 Report
The review does address the Clinical Utility of lncRNAs and their Application as Molecular biomarkers in Breast Cancer. The language is well understood, but please review the punctuation in the text.
It is interesting what is included in this review for the calculation of the sample size for the discovery of biomarkers in clinical research, and the determination of the sample size for lncRNA studies in breast cancer research, which could be very applicable to other studies in other types of cancer
However, I would suggest the following:
In the abstract: The authors mention... in particular, the lncRNA DSCAM-AS1 and the GATA3-AS1 which serve as examples due to their high subtype-specific expression profile in luminal type B breast cancer, however, ¿the authors are proposing these examples as the most proper for the future diagnosis? or are they just examples? generally in the abstracts examples are not necessary, only when these are relevant to the study.
authors could deeper about other lncRNA's breast cancer-relevant functions toward favoring its capacity for proliferation and survival
The authors must review the organization of the information because after the introduction the information could be organized into sections
The authors should include how they obtained the information for this review, for example, The data collection process
Figure 1 shows the workflow for the validation of lncRNAs as biomarkers with utility and clinical application, however, this image could be improved, I would recommend changing the position of the flow to have more space for internal photos, and thus a way to have a better quality image in resolution and information
Author Response
Response to Reviewer 3 Comments
The review does address the Clinical Utility of lncRNAs and their Application as Molecular biomarkers in Breast Cancer. The language is well understood, but please review the punctuation in the text.
It is interesting what is included in this review for the calculation of the sample size for the discovery of biomarkers in clinical research, and the determination of the sample size for lncRNA studies in breast cancer research, which could be very applicable to other studies in other types of cancer
However, I would suggest the following:
Point 1: In the abstract: The authors mention... in particular, the lncRNA DSCAM-AS1 and the GATA3-AS1 which serve as examples due to their high subtype-specific expression profile in luminal type B breast cancer, however, ¿the authors are proposing these examples as the most proper for the future diagnosis? or are they just examples? generally in the abstracts examples are not necessary, only when these are relevant to the study.
Response 1: Thank you very much for the comment. The lncRNAs DSCAM-AS1 and GATA3-AS1 were considered as examples of important lncRNAs in molecular diagnostics. In particular, these transcripts are mentioned in the abstract because they are clear examples of lncRNAs that may have relevance in molecular diagnosis of breast cancer over luminal tumors in the near future.
Point 2: Authors could deeper about other lncRNA's breast cancer-relevant functions toward favoring its capacity for proliferation and survival.
Response 2: Thank you very much for the comment, we agree. In the manuscript, we integrated information about the relevant functions of some lncRNAs discussed in the review. We mentioned their main function in cancer, and whether they were associated with any cellular process as suggested by reviewer 3. Please see lines 38, 45, 54, 213-215, and 250-251. Furthermore, a column including the biological functions of each lncRNA was included in Table 1. Please see lines 406-407, 417, and 419-426.
Point 3: The authors must review the organization of the information because after the introduction the information could be organized into sections
Response 3: Thank you very much for the comment. All the information in the manuscript was organized by sections. After the introduction, the topics included were the following:
- Introduction (line, 36).
- Sample size calculation for biomarker discovery in clinical research (line, 80).
2.1. Sample size determination for lncRNA studies in breast cancer research (line, 155).
- The current use of lncRNAs as clinical biomarkers in clinical practice (line, 175).
3.1. Challenges and perspectives for lncRNA clinical application as predictive biomarkers for breast cancer management (line, 231).
3.2. The use of lncRNAs as molecular biomarkers in the RNA-Based Therapeutics Era (line, 270).
3.3. The current challenges for lncRNA research and for their implementation as molecular biomarkers in routine clinical practice (line, 368).
In each of the sections, a particular problem is addressed, and each section is described in detail, citing literature related to each one, and ending with a final conclusion in each paragraph emphasizing the described information.
Point 4: The authors should include how they obtained the information for this review, for example, The data collection process.
Response 4: Thank you very much for your comment, we agree. The final part of the introduction mentions how the systematic search was carried out. Please see lines 148-154.
Point 5: Figure 1 shows the workflow for the validation of lncRNAs as biomarkers with utility and clinical application, however, this image could be improved, I would recommend changing the position of the flow to have more space for internal photos, and thus a way to have a better quality image in resolution and information.
Response 5: Thank you very much for the comment, we agree. Figure 1 was modified according to the suggestions made by reviewer 3. Please see line 331. Finally, the punctuation in the manuscript was revised, as suggested by reviewer 3.
In the hope that we have responded to all the suggestions made by the Reviewer#3, we sincerely hope that the new modifications made to the manuscript are in accordance with what was requested. Thank you for the time invested in reviewing our work.
